# Biogas Reforming over Al-Co Catalyst Prepared by Solution Combustion Synthesis Method

Manapkhan Zhumabek [1,2], Galina Xanthopoulou [3], Svetlana A. Tungatarova [1,4,*],
Tolkyn S. Baizhumanova [1,4], George Vekinis [3] and Dmitry Yu. Murzin [5]

1   Laboratory of Oxidative Catalysis, D.V. Sokolsky Institute of Fuel, Catalysis and Electrochemistry, 142, Kunaev Str., Almaty 050010, Kazakhstan; manapkhan_86@mail.ru (M.Z.); baizhuma@mail.ru (T.S.B.)
2   Institute of Chemical and Biological Technologies, Satbayev University, 22a, Satpaev Str., Almaty 050013, Kazakhstan
3   Institute of Nanoscience and Nanotechnology, NCSR Demokritos, 15310 Athens, Greece; g.xanthopoulou@inn.demokritos.gr (G.X.); g.vekinis@inn.demokritos.gr (G.V.)
4   Chemistry and Chemical Technologies Faculty, Al-Farabi Kazakh National University, 71, Al-Farabi Str., Almaty 050040, Kazakhstan
5   Johan Gadolin Process Chemistry Centre, Åbo Akademi University, 20500 Åbo (Turku), Finland; dmitry.murzin@abo.fi
*   Correspondence: tungatarova58@mail.ru; Tel.: +7-727-291-6632

**Abstract:** The results of carbon dioxide reforming of $CH_4$ (model biogas) on catalysts prepared by solution combustion synthesis (SCS) and impregnation of moisture capacity methods are presented. Investigation of the activity of catalysts synthesized from initial mixtures of $Co(NO_3)_2$-$Al(NO_3)_3$-urea of different compositions was carried out for the production of synthesis-gas, and SCS and traditional incipient wetness impregnation catalyst preparation methods were compared. The methane conversion reached 100%, and the conversion of $CO_2$ increased to 86.2%, while the yield of $H_2$ and CO was 99.2% and 85.4%, respectively, at 900 °C. It was found that $CoAl_2O_4$ spinel formation was due to substitution of $Al^{3+}$ with $Co^{2+}$ cations. Consequently, $CoAl_2O_4$ lattice parameters increased, since the ionic radius of $Al^{3+}$ (0.51 Å) less than $Co^{2+}$ (0.72 Å). Advantages of SCS catalysts in comparison with catalysts prepared by the traditional incipient wetness impregnation method in dry reforming of methane were shown. The aim of this work is to develop a new catalyst for the conversion of model biogas into synthesis gas, which will contribute to the organization of a new environmentally friendly, energy-saving production in the future.

**Keywords:** biogas; synthesis gas; reforming; composite materials



## 1. Introduction

In recent years, an intensive study of the processes involving biogas as an alternative source for not only energy, but also as a raw material for the production of petrochemicals, started throughout the world due to the inevitable exhaustion of non-renewable resources for energy generation and petrochemical production. Biogas resulting from the anaerobic fermentation of biomass and from any organic waste is a practically inexhaustible renewable resource for obtaining valuable products such as synthesis gas, hydrogen, and hydrocarbons. Even biogas of "low" quality is suitable for processing into valuable raw materials for power engineering and petrochemistry, making it possible to avoid expensive methods of production.

Methane and $CO_2$ are major components of biogas. Development of the scientific foundations of its use as a raw material for the production of liquid motor fuels and a number of other products seems to be an urgent task, since biogas is a renewable raw material [1–4]. Currently, biogas is mainly used only as a fuel. Therefore, practical use is mainly limited to building installations in rural areas for heating various rooms, greenhouses, warehouses, etc. Chemical processing of biogas into synthesis gas is the

most promising option for its utilization [5–7]. Synthesis gas is a modern environmentally friendly fuel that burns without harmful impurities. On the other hand, synthesis gas is the basis of petrochemical production. Today, the leading oil companies in the world are engaged in the selection of conditions for the production of synthesis gas. Biogas is a very convenient component for its synthesis, since it contains both $CH_4$ and $CO_2$ reaction components.

In the coming years, biogas obtained through processing of organic wastes will probably be the only solution to the problems of energy supply to agricultural enterprises. In the future, with the exhaustion of natural hydrocarbons, this direction may be the only alternative way to obtain both motor fuel and petrochemicals. Extensive resources, high profitability of production, and favorable environmental properties make biogas the most promising source of hydrocarbons, capable of meeting the current and future needs of humanity for energy and hydrocarbons. The catalytic processing of biogas into synthesis gas opens up fundamentally new opportunities for creating relatively simple and low-tonnage gas chemical technologies.

This work complies with the principles of the green economy: (i) the use of renewable raw materials, i.e., biogas; and (ii) the involvement of greenhouse gases, i.e., methane and carbon dioxide.

The aim of this work is to develop new catalysts for the conversion of model biogas (a mixture of methane with carbon dioxide) into synthesis gas, which will contribute to the organization of new environmentally friendly, energy-saving production in the future.

Bulk metals [8,9] and oxides [10,11], as well as supported on various types of carriers [12], are used as catalysts for the carbon dioxide conversion of methane. However, in recent years, new types of highly active catalysts made in the combustion process have appeared. Self-propagating high-temperature synthesis (SHS) [13] and, in particular, its modern modification—solution combustion synthesis (SCS) [14–16]—is a new method for obtaining a modern class of catalysts of various uses based on metals, alloys, oxides, spinels, etc. The mode of a strong exothermic reaction (i.e., combustion reaction), in which the heat release is localized in the layer and transferred from layer to layer by heat transfer, is carried out in the SHS process. Structures with a high concentration of defects, which is one of the reasons for the high activity of SHS catalysts, are formed under conditions of rapid rates of combustion and cooling reactions. Studies of the physicochemical and mechanical properties of a wide range of synthesized SHS and SCS catalysts have been reported in the literature [17–22]. As a result of these studies, SHS materials with high catalytic activity—promising for many industrial processes such as partial oxidation, reduction, carbon dioxide conversion of methane, etc.—have been developed [23–25]. In this study, a series of catalysts based on Co and Al was synthesized by SCS and incipient wetness impregnation methods, and characterized by a range of physicochemical methods. The catalysts were investigated in dry methane reforming in a continuous fixed bed reactor.

## 2. Results

### 2.1. Characterization of Catalysts

The following reactions are possible in the process of solution combustion synthesis (Table 1).

**Table 1.** Solution combustion reactions of the $Co(NO_3)_2 + Al(NO_3)_3 + H_2O +$ urea system.

| Reactions | Remarks |
|---|---|
| $2Al(NO_3)_3 + 5CH_4N_2O + Co(NO_3)_2 \rightarrow Co + CoO + Co_xAl_y +$ $Co_xAl_{2-x}O_3 + 5CO_2 + 8N_2 + 10H_2O$ | 500 °C. The total reaction |
| $Al_2O_3 + C \rightarrow Al + CO_2, CoO + C \rightarrow Co$ | Carbon is formed by burning urea and reduces oxides to metal |
| $Al+O_2 \rightarrow Al_2O_3$ | $\Delta H^\circ_{278} = -3352$ kJ/mol, exothermic reaction. The reduced metals can partially oxidize or react with another metal |
| $Co+O_2 \rightarrow CoO$ | $\Delta H^\circ_{278} = -475.8$ kJ/mol, exothermic reaction |
| $Al_2O_3 + CoO \rightarrow CoAl_2O_4, Al_2O_3 + CoO \rightarrow Co_2AlO_4$ | Synthesis of spinel, endothermic reaction |
| $Co + Al \rightarrow Co_xAl_y$ | Synthesis of intermetallic compound |

The composition of the initial mixture, combustion conditions, and the final catalyst compositions are shown in Table 2.

**Table 2.** The initial compositions of salts and final catalyst composition at 500 °C preheating temperature of solution.

| Starting Compounds | Catalysts Composition |
| --- | --- |
| 60% Co(NO$_3$)$_2$ + 40% Al(NO$_3$)$_3$ + 60% urea + 3 mL H$_2$O | CoAl$_2$O$_4$, Co$_2$AlO$_4$, Co$_3$O$_4$, CoO, Al, AlCo |
| 50% Co(NO$_3$)$_2$ + 50% Al(NO$_3$)$_3$ + 60% urea + 3 mL H$_2$O | CoAl$_2$O$_4$, Co$_2$AlO$_4$, Co$_3$O$_4$, CoO, Al, AlCo |
| 40% Co(NO$_3$)$_2$ + 60% Al(NO$_3$)$_3$ + 60% urea + 3 mL H$_2$O | CoAl$_2$O$_4$, Co$_2$AlO$_4$, Co$_3$O$_4$, CoO, Al, AlCo |
| 30% Co(NO$_3$)$_2$ + 70% Al(NO$_3$)$_3$ + 60% urea + 3 mL H$_2$O | CoAl$_2$O$_4$, Co$_2$AlO$_4$, Co$_3$O$_4$, CoO, Al, AlCo |

### 2.1.1. XRD Analysis

The resulting catalysts had similar qualitative compositions but differed in their phase ratios. The approximate ratio between the phases was determined from the relative intensities of the X-ray diffraction peaks for each phase (Figure 1).

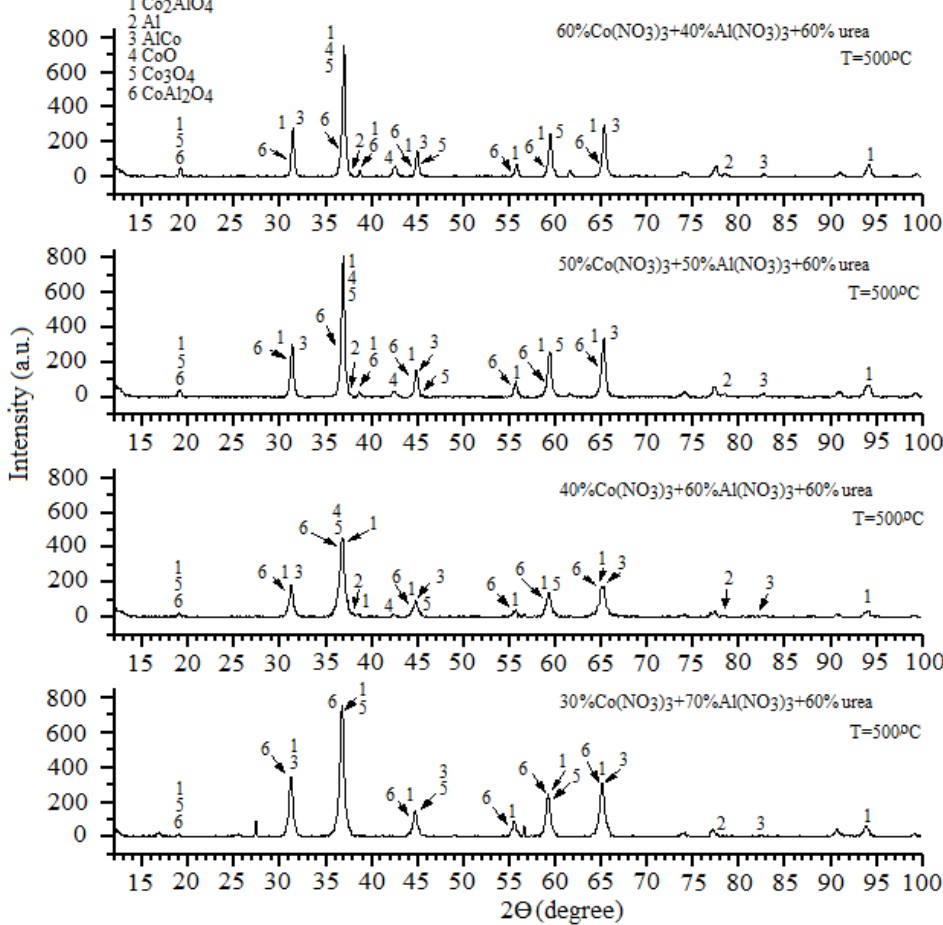

**Figure 1.** X-ray spectra for the Co(NO$_3$)$_2$ + Al(NO$_3$)$_3$ + urea + H$_2$O system at a preheating temperature of the initial mixture of 500 °C.

Temperature curves measured during SCS showed a second peak after SCS (Figure 2). This could only be explained by the reaction of carbon with metal oxides. Only reaction Al$_2$O$_3$ + C → Al + CO$_2$ could explain presence of Al in the product of reaction because hydrogen, which appears during the reaction, cannot reduce Al$_2$O$_3$ under conditions of synthesis.

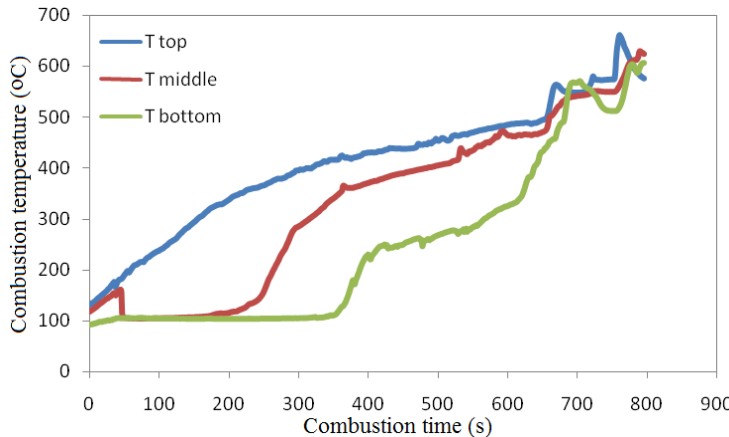

**Figure 2.** Temperature curves during SCS of sample with initial batch 30% Co(NO$_3$)$_2$ + 70% Al(NO$_3$)$_3$ + 60% urea.

Increase in the concentration of Al(NO$_3$)$_3$ in the solution led to an increase in the concentration of both Co$_2$AlO$_4$ and CoAl$_2$O$_4$ spinels in the final product (Figure 3a,c). The concentrations of intermetallic compounds were several times lower than that of spinels, and practically did not affect the general pattern of increasing spinel concentration with increasing Al(NO$_3$)$_3$ in the initial mixture (Figure 3b,d). As shown in Figure 3a,b, the concentrations of spinels were significantly higher than those of CoO and were comparable to those of Co$_3$O$_4$.

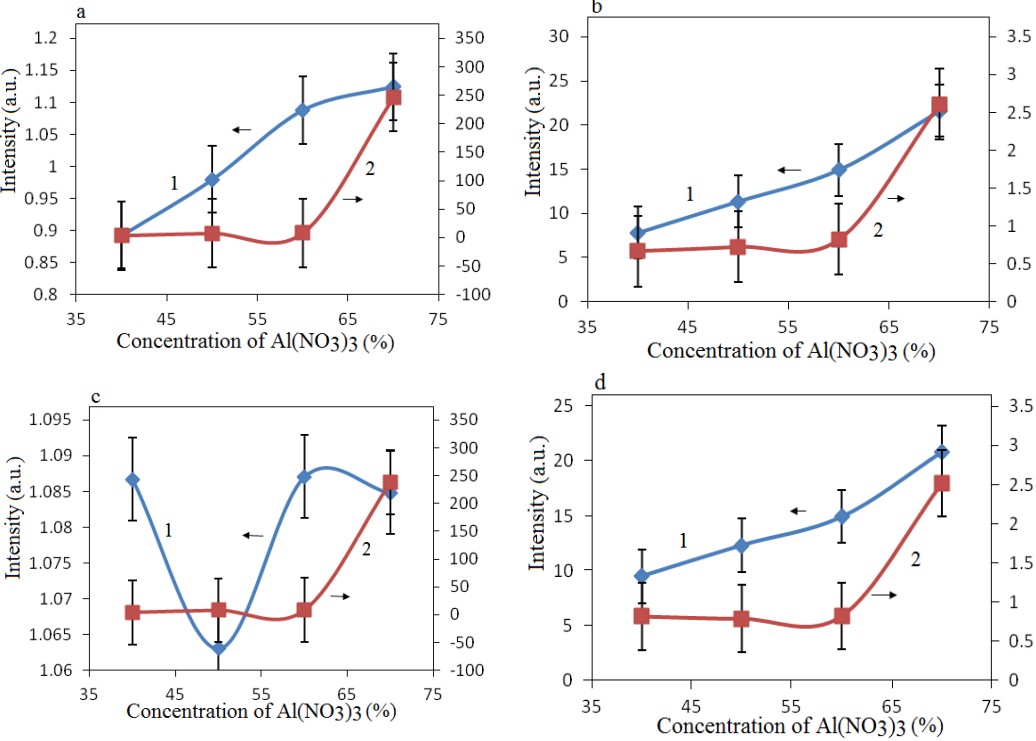

**Figure 3.** Change in the intensity of XRD peaks depending on the concentration of Al(NO$_3$)$_3$ in the initial mixture. (**a**) 1—CoAl$_2$O$_4$/Co$_3$O$_4$, 2—CoAl$_2$O$_4$/CoO; (**b**) 1—CoAl$_2$O$_4$/Al, 2—CoAl$_2$O$_4$/AlCo; (**c**) 1—Co$_2$AlO$_4$/Co$_3$O$_4$, 2—Co$_2$AlO$_4$/CoO; (**d**) 1—Co$_2$AlO$_4$/Al, 2—Co$_2$AlO$_4$/AlCo.

The concentration of Al(NO$_3$)$_3$ played an important role in the deformation of the crystal lattice. The higher the concentration of Al(NO$_3$)$_3$ in the initial solution, the more Co$^{2+}$ ions (0.72 Å) could replace the Al$^{3+}$ ions (0.51 Å) in the matrix even though, generally,

$Co^{2+}$ cations occupy voids in the $Al_2O_3$ matrix, something which was not observed by XRD. This was illustrated by the observed increase in the size of the crystal lattice of spinel and cobalt oxides. No structural changes of intermetallic compounds and aluminum were observed (Figure 4).

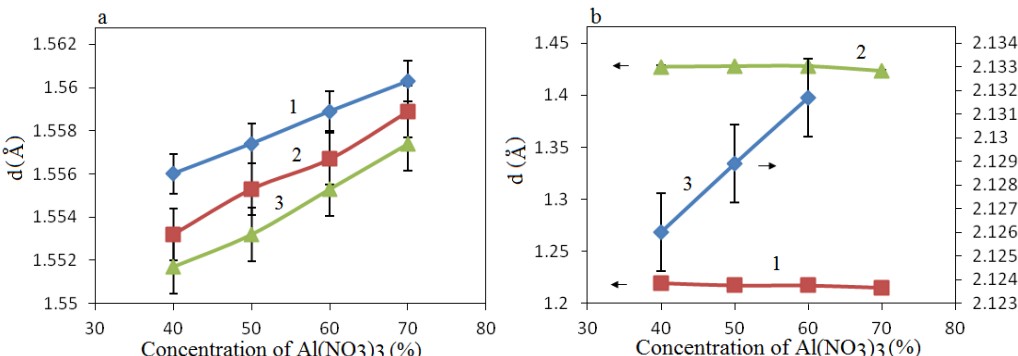

**Figure 4.** Effect of the concentration of $Al(NO_3)_3$ in the initial mixture on the size of the crystal lattice for $CoAl_2O_4$, $Co_2AlO_4$, $Co_3O_4$, $CoO$, $AlCo$, $Al$: (**a**) 1—$CoAl_2O_4$ (hkl = 511), 2—$Co_2AlO_4$ (hkl = 511), 3—$Co_3O_4$ (hkl = 511); (**b**) 1—$Al$ (hkl = 311), 2—$AlCo$ (hkl = 200), 3—$CoO$ (hkl = 200).

The average crystallite size of the obtained catalysts was estimated from the width of the XRD peaks. The results presented in Figure 5 show that the average particle size decreased with increasing $Al(NO_3)_3$ content.

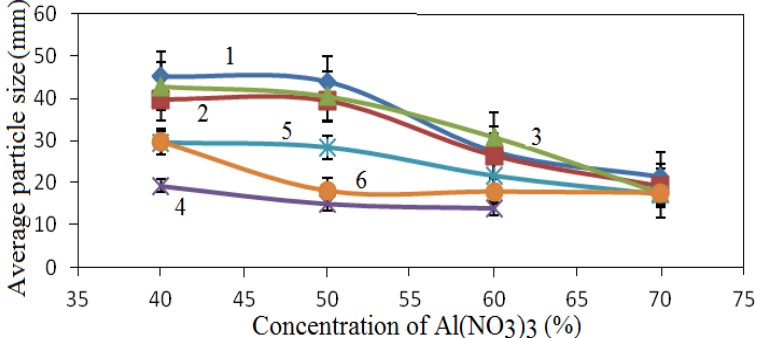

**Figure 5.** Dependence of the crystallite size on the concentration of $Al(NO_3)_3$; 1—$CoAl_2O_4$ (511), 2—$Co_2AlO_4$ (511), 3—$Co_3O_4$ (511), 4—$CoO$ (200), 5—$AlCo$ (200), 6—$Al$ (311).

The surface area of all obtained catalysts was measured by the BET method. The areas were the same (33.1–33.4 $m^2g^{-1}$).

### 2.1.2. SEM Analysis

The synthesized catalysts were examined with scanning electron microscopy to study the surface structure and morphology. In particular, catalysts containing 30% and 60% $Co(NO_3)_2$ in the initial mixture, synthesized during preheating to 500 °C, were studied (Figures 6 and 7). Nanosized particles were determined in catalysts prepared by SCS. An EDS analysis was also conducted to determine the composition, which comprised $CoAl_2O_4$, $Co_2AlO_4$ (cubic), $Co_xAl_y$ (cubic), $Co_3O_4$, $Al$, and $CoO$ (cubic).

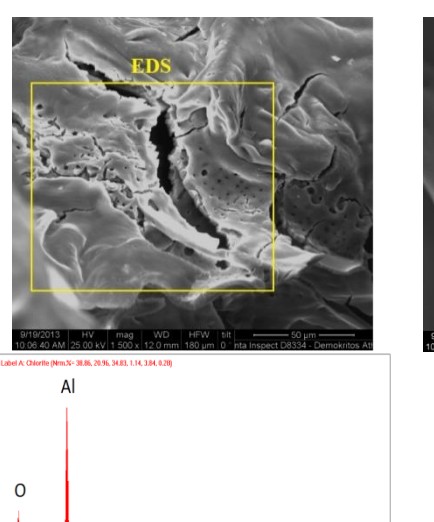

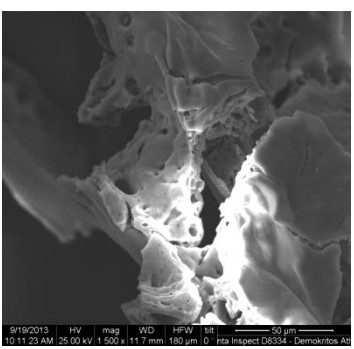

**Figure 6.** SEM images of the catalyst with the initial composition of 30% $Co(NO_3)_2$ + 70% $Al(NO_3)_3$ at T = 500 °C and EDS of the catalyst at the point indicated in the photo.

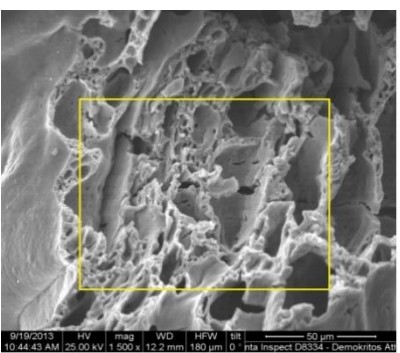

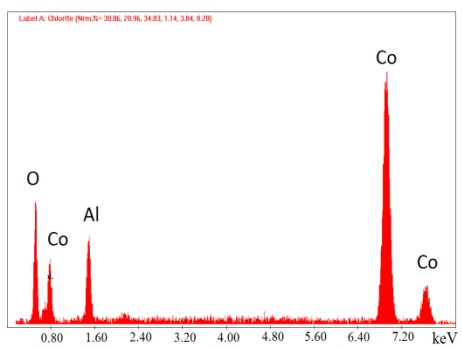

**Figure 7.** SEM image of the catalyst with the initial composition of batch 60% $Co(NO_3)_2$ + 40% $Al(NO_3)_3$ at T = 500 °C and EDS of catalyst at the point indicated in the photo.

## 2.2. Catalytic Results

Catalysts obtained by solution combustion synthesis were tested in dry reforming of methane with carbon dioxide in the temperature range from 750 to 900 °C. Data on the conversion of $CH_4$ and $CO_2$, as well as the $H_2/CO$ ratio in the reaction products for the most active catalysts prepared from 60% $Co(NO_3)_2$ + 40% $Al(NO_3)_3$ and 30% $Co(NO_3)_2$ + 70% $Al(NO_3)_3$ mixtures are shown in Figure 8.

The results show that 100% $CH_4$ conversion was observed on a catalyst containing 30% $Co(NO_3)_2$ + 70% $Al(NO_3)_3$ at the highest temperature. This catalyst also showed 100% conversion of $CO_2$. The $H_2/CO$ ratio of the reaction products was 0.8–1.2 for both catalysts. An increase in the reaction temperature led to an increase in the $H_2/CO$ ratio. The 1:1 ratio is very important for the production of dimethyl ether, which can be used as a fuel. Conversion of $CH_4$ was also 100% at 900 °C in the presence of the 60% $Co(NO_3)_2$ + 40% $Al(NO_3)_3$ catalyst, while $CO_2$ conversion reached only 86.2%. The yield of $H_2$ increased to 99.2% and yield of CO was 85.4%" instead of "The yields of $H_2$ and CO were 99.2% and 85.4%, respectively (Figure 8). Tests on the 30% $Co(NO_3)_2$ + 70% $Al(NO_3)_3$ catalyst showed that the maximum 100% conversion of $CH_4$ and $CO_2$ were achieved. The yields reached 99.2% $H_2$ and 99.1% CO at 900 °C (Figure 8).

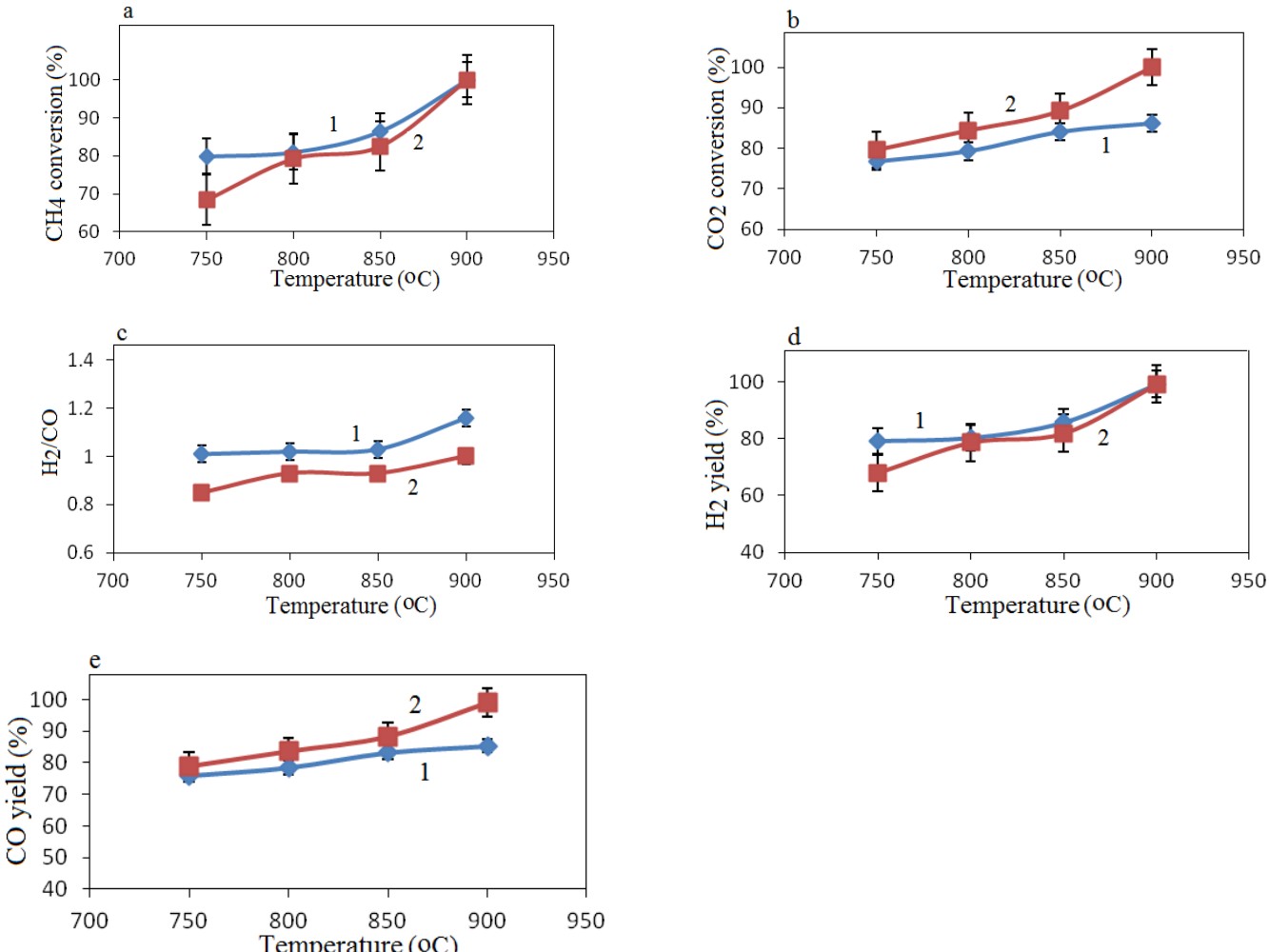

**Figure 8.** Conversion of $CH_4$ (**a**) and $CO_2$ (**b**), $H_2/CO$ ratio (**c**), $H_2$ yield (**d**) and CO (**e**) for SCS co-catalysts depending on the reaction temperature of the carbon dioxide conversion of methane; 1: 60% $Co(NO_3)_2$ + 40% $Al(NO_3)_3$ + 60% urea; 2: 30% $Co(NO_3)_2$ + 70% $Al(NO_3)_3$ + 60% urea. GHSV = 860 $h^{-1}$.

The effect of space velocity was more important for the conversion of $CH_4$ than for $CO_2$ conversion (Figure 9a), and because methane was the source of hydrogen in this reaction, the $H_2/CO$ ratio decreased with increasing space velocity (Figure 9b). This phenomenon can be explained by the fact that $CH_4$ molecules (3.99 Å) are almost twice as large as $CO_2$ molecules (2.3 Å). Therefore, the adsorption and desorption of $CH_4$ molecules is more limited than that of $CO_2$ molecules under reaction conditions at high space velocity.

Stability of the catalyst was also investigated (initial composition of 30% $Co(NO_3)_2$ + 70% $Al(NO_3)_3$) at GHSV 3300 $h^{-1}$ (Figure 10) at which the highest catalytic activity was observed (Figure 9). It should be noted that the catalyst retained high catalytic activity at the end of 7 h: 90% $CH_4$ conversion and 94% $CO_2$ conversion (Figure 10a), 89% $H_2$ yield and 93% CO yield (Figure 10b), and the $H_2/CO$ ratio was 1 (Figure 10a).

A study of the activity of 30% $Co(NO_3)_2$ + 70% $Al(NO_3)_3$ + urea catalysts prepared by SHS and incipient wetness impregnation methods in dry reforming of methane was carried out at 850 and 900 °C. The results on the yield of hydrogen and CO for these catalysts at a space velocity of 3300 $h^{-1}$ are presented in Figure 11.

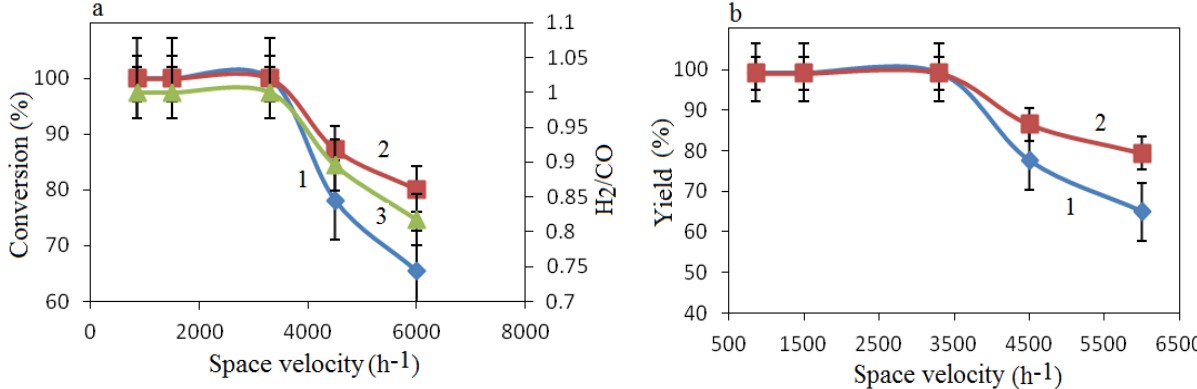

**Figure 9.** Effect of space velocity on conversion of $CH_4$, $CO_2$ and $H_2/CO$ ratio at 900 °C (**a**) in reaction products, and $H_2$ and CO yields (**b**). (**a**) 1—$CH_4$ conversion, 2—$CO_2$ conversion, 3—$H_2/CO$ ratio; (**b**) 1—$H_2$ yield, 2—CO yield.

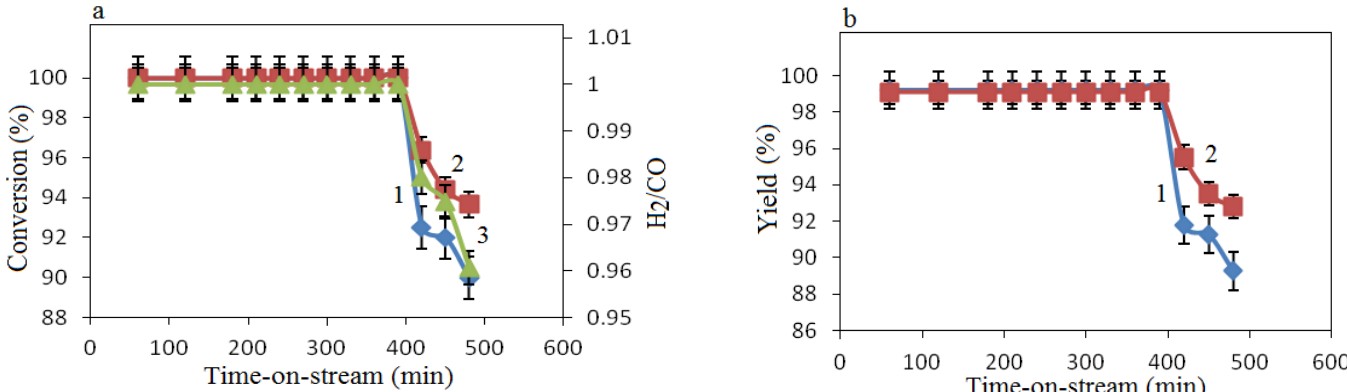

**Figure 10.** Dependence of the conversion of $CH_4$, $CO_2$, $H_2/CO$ ratio and yields of products vs. time-on-stream. (**a**) 1—$CH_4$ conversion, 2—$CO_2$ conversion, 3—$H_2/CO$ ratio; (**b**) 1—$H_2$ yield, 2—CO yield; GHSV = 3300 $h^{-1}$.

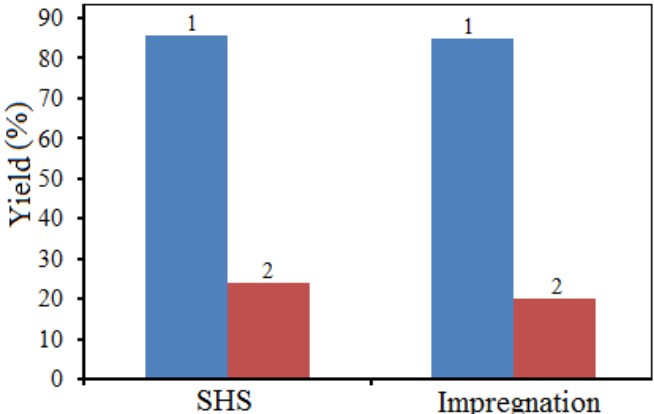

**Figure 11.** Effect of the catalyst preparation conditions on the yield of $H_2$ and CO at 900 °C for the 30% $Co(NO_3)_2$ + 70% $Al(NO_3)_3$ + 60% urea catalyst; 1—yield of $H_2$, 2—yield of CO, GHSV-3300 $h^{-1}$.

It was found that the yields of $H_2$ as well as the yields of CO for the compared processes were quite close, with a slight advantage of the SCS method when comparing the data on carbon dioxide conversion of $CH_4$ on the 30% $Co(NO_3)_2$ + 70% $Al(NO_3)_3$ + 60% urea catalysts prepared by the SCS method and incipient wetness impregnation. The yield of $H_2$ varied between 85% (incipient wetness impregnation) and 87% (SCS), while the yield of CO varied between 20% (incipient wetness impregnation) and 24% (SCS). The

effect of addition of water vapor to the initial reaction mixture was shown in the process of carbon dioxide conversion of $CH_4$ on the above catalyst prepared by SCS method at a space velocity of $2500\ h^{-1}$. Figure 12a,b shows the effect of water vapor on the yield and selectivity of the process.

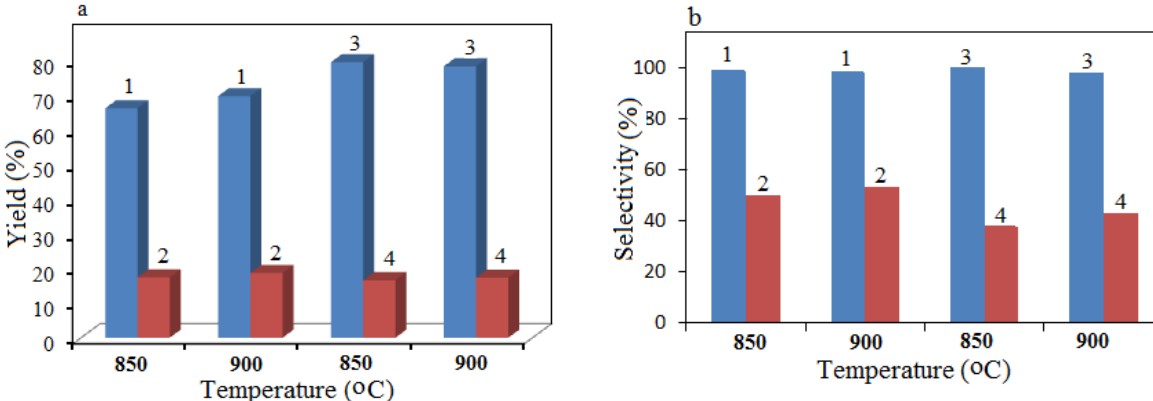

**Figure 12.** Dependence of the yield (**a**) and selectivity (**b**) on the addition of water vapor to reaction mixture; 1—$H_2$, 2—CO without water vapor; 3—$H_2$, 4—CO in the presence of water vapor.

As a result of varying the temperature, it was found that carrying out the process at 900 °C in the presence of water vapor made it possible to achieve the highest values of both yield and selectivity for $H_2$ and CO. The presence of water vapor had a significant effect on yields of products, while the effect on selectivity was small. It was also shown that water vapor did not have any effect on the catalysts prepared by the impregnation method. When comparing the SCS and impregnation methods at a space velocity of $2500\ h^{-1}$, the advantage of the SCS method was approximately 7–9%.

## 3. Discussion

Since the $CoAl_2O_4$ spinel is the main catalytic component, the $CH_4$ conversion depends on the parameters of the crystal lattice of this spinel. In $CH_4$, the C–H bond length, which must be broken in accordance with the reaction mechanism, is 1.54 Å. In the spinel, the size of the crystal lattice varies from 1.559 to 1.561 Å. Changes in the parameters of the crystal lattice occur due to an increase in the Al content in the crystal lattice of cobalt oxide. These parameters are very close to the C–H parameter. Therefore, 60% $Co(NO_3)_2$ + 40% $Al(NO_3)_3$ and 30% $Co(NO_3)_2$ + 70% $Al(NO_3)_3$ catalysts demonstrate high activity.

As for the conversion of $CO_2$, the length of the C–O bond in $CO_2$ is 1.43 Å. $Co_xAl_y$, which is present in these catalysts, has the closest lattice parameters. Higher conversions are observed on the 30% $Co(NO_3)_2$ + 70% $Al(NO_3)_3$ catalyst than on the 60% $Co(NO_3)_2$ + 40% $Al(NO_3)_3$ sample (Figure 8). Probably, this is due to the lattice parameter of 1.423 Å of the $Co_xAl_y$ phase in the 30% $Co(NO_3)_2$ + 70% $Al(NO_3)_3$ catalyst, which is close to the required 1.43 Å. Thus, activation of the C–O bond presumably takes place on $Co_xAl_y$.

Composition of the catalysts was examined by XRD after each experiment (Figure 13). Mass of the 60% $Co(NO_3)_2$ + 40% $Al(NO_3)_3$ + 60% urea and 30% $Co(NO_3)_2$ + 70% $Al(NO_3)_3$ + 60% urea catalysts after the catalytic reaction decreased by 3.9% and 11.11%, respectively. It follows from the X-ray spectrum that carbon is formed after the experiments in addition to cobalt carbide, $CoC_x$. In this case, the cobalt oxide disappears after the experiments, concomitant with the appearance of cobalt. Cobalt oxide is thus reduced to Co. This reaction causes a decrease in the weight of the catalyst after experiments.

Coking of the studied catalysts used for dry methane reforming was detected by SEM/EDX. The results presented in Figures 14 and 15 show that the catalyst was carbonized after 8 h of operation and that the carbon fibers had a multichannel nano structure (Figure 16).

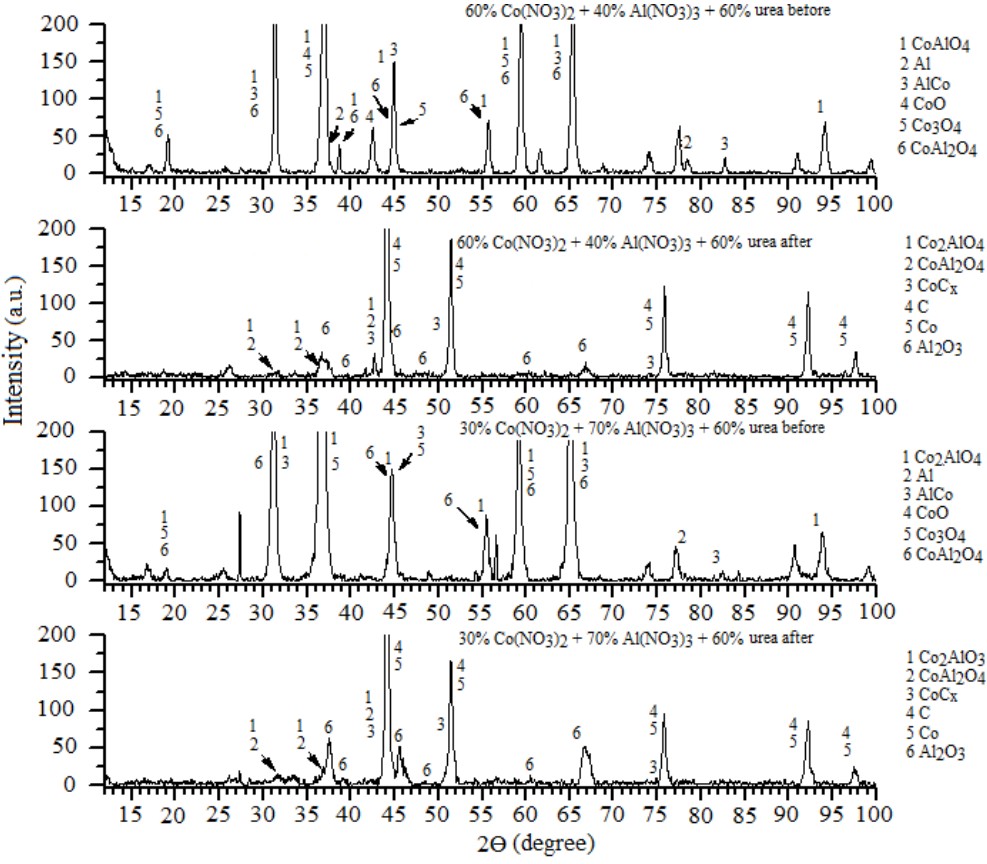

**Figure 13.** X-ray spectra for catalysts before and after the reaction.

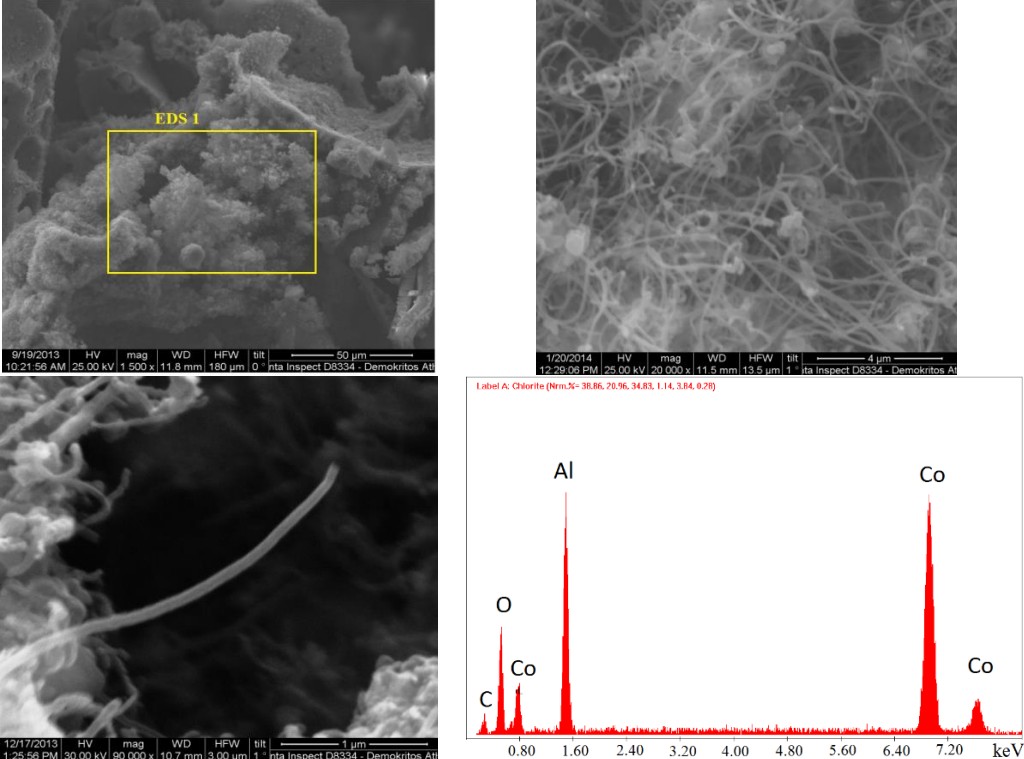

**Figure 14.** SEM images of the 30% Co(NO$_3$)$_2$ + 70% Al(NO$_3$)$_3$ catalyst (synthesis temperature T = 500 °C) after 8 h of operation, and EDS at the point marked in the image.

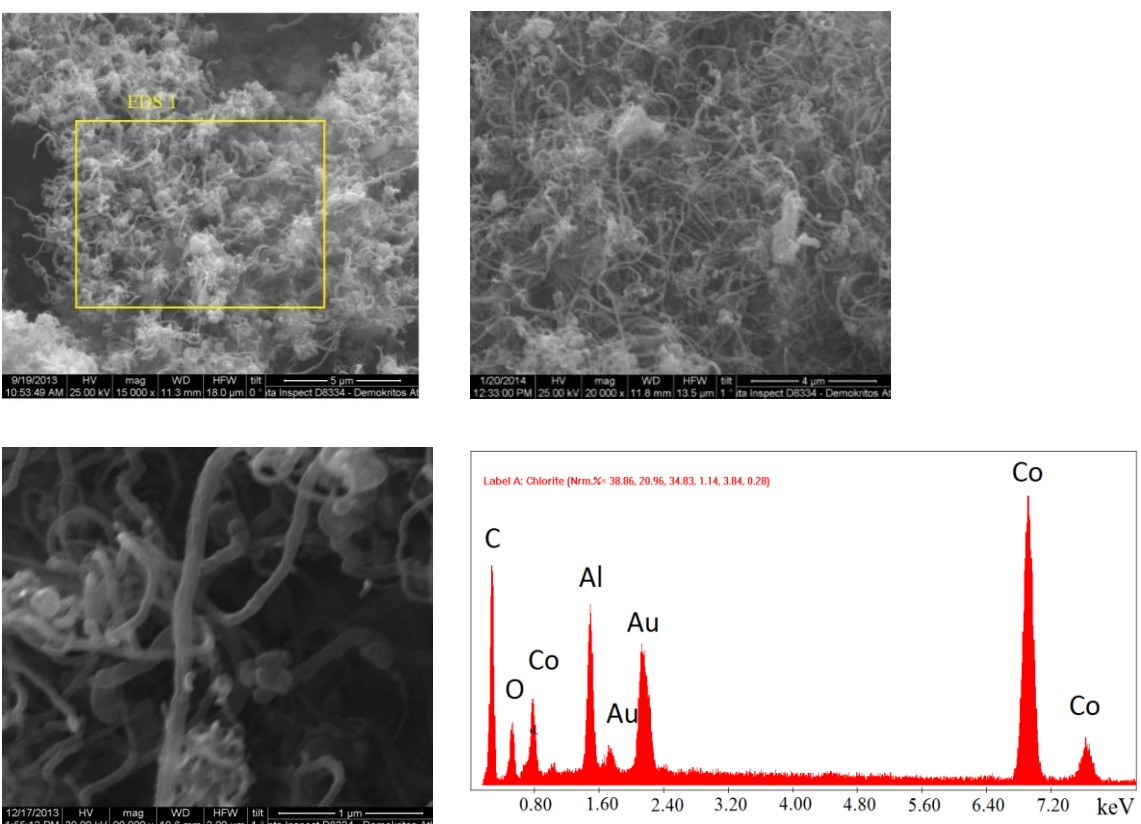

**Figure 15.** SEM images of the 60% Co(NO$_3$)$_2$ + 40% Al(NO$_3$)$_3$ catalyst (synthesis temperature T = 500 °C) after 8 h of operation, and EDS at the point marked in the image.

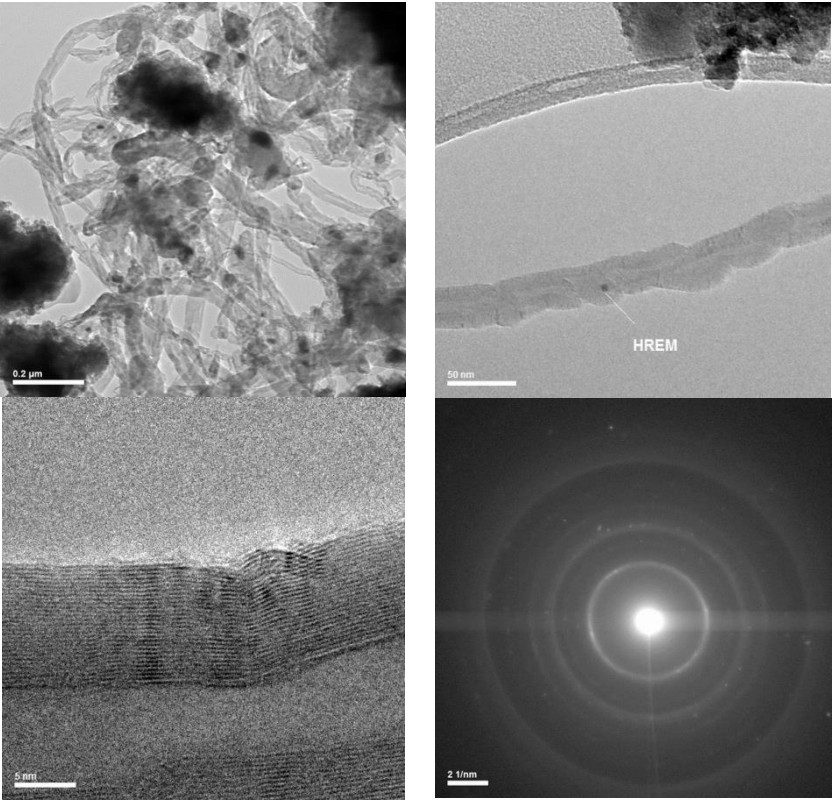

**Figure 16.** TEM images of the carbon fibers on the catalyst, which worked for 8 h.

## 4. Materials and Methods

Catalysts based on Co and Al were prepared by the SHS method in solution–solution combustion synthesis. For the preparation of catalysts, pre-calculated amounts of nitrate salts were used: $Co(NO_3)_2 \times 6H_2O$ (98–99% Sigma, Aldrich, St. Louis, MO, USA), $Al(NO_3)_3 \times 9H_2O$ (98–99% Sigma, Aldrich), and urea (98%, LLP Labhimprom, Kazakhstan). These salts were pre-ground in an agate mortar and then mixed in a porcelain cup. Thereafter, 10 mL of distilled water was gradually added to this mixture of salts. The resulting mixture was stirred in air for several minutes until complete dissolution. A muffle furnace was preheated to the required temperature (up to 500 °C). The prepared mixture was transferred from a porcelain cup to a heat-resistant glass beaker with a volume of 200 mL and placed in the heated muffle furnace. After 2–3 min, when the muffle furnace door was not fully opened, it was visually possible to observe combustion in the solution, in which this mixture rose over the walls of the glass when boiling. Urea was added to the composition of SCS catalysts to improve the combustion process. The presence of urea in the catalyst promoted a change of the solution to a brown color during combustion. The beaker was then cooled in air, and the prepared catalyst was placed in a glass jug.

The second series of catalysts was prepared by the incipient wetness impregnation of dispersed α-$Al_2O_3$ (granule size 150–200 μm, S–57.7 $m^2g^{-1}$) by water solutions of metal nitrates with subsequent heating in air at 250 °C for 5 h, at 600 °C for 2 h, and at 900 °C for 1 h.

Temperature curves were measured during SCS in the muffle furnace heated to 500 °C. Three thermocouples were installed on top of the muffle furnace. All thermocouples were inside the glass. Thermocouples were in contact with the bottom, middle, and top layers of the solution. Two combustion modes—a volumetric explosion and a self-propagating mode—occurred during the synthesis of catalysts by solution combustion. The solution was heated, and water evaporated in the volumetric explosion mode. The gel formed after evaporation of water. The temperature in the muffle furnace gradually increased to critical. As soon as the temperature reached a critical value, the exothermic reaction took place throughout the entire volume of the catalyst.

Atomic structures of the catalysts were determined by X-ray diffraction measurements on a Siemens Spellman DF3 spectrometer with Cu-Kα radiation. KCl (10%) was added to samples as an internal standard to allow for a semi-quantitative XRD analysis. Brunauer–Emmett–Teller (BET) specific surface area was measured on a GAPP V-Sorb 2800 Analyzer using nitrogen as a carrier gas. In the process of solution combustion synthesis, the nanopowders were obtained; therefore, the porosity of them was not measured. The microstructures of the materials were examined after spatter coating with gold (coating thickness 5–10 nm) by a scanning electron microscope (Quanta Inspect from FEI) together with point EDX elemental analysis.

Experiments on the partial oxidation of methane to synthesis gas were carried out on a flow-type installation at atmospheric pressure in a tubular quartz reactor with a fixed catalyst bed without any pre-reduction. The catalyst was placed in the central part of the reactor, and quartz wool was placed above and below the catalyst bed. The catalytic reaction was carried out at 750, 800, 850, and 900 °C using a mixture of $CO_2$:$CH_4$ in the ratio of 1:1 as the feed. The basic dry reforming of the methane reaction is

$$CO_2 + CH_4 \rightarrow 2CO + 2H_2 \tag{1}$$

In addition to the reforming process, by-products were formed by the following reactions:

$$CH_4 + CO_2 \rightarrow 2C + 2H_2O \tag{2}$$

$$CO_2 + CH_4 \rightarrow CO + C_xH_y \tag{3}$$

Such by-products were observed for the majority of catalysts in very small amounts, indicating very high selectivity for CO and $H_2$.

Analyses of the initial mixture and reaction products were carried out using a chromatograph "Chromos GC-1000" with the "Chromos" software and on a chromatograph "Agilent Technologies 6890N" (Santa Clara, CA, USA) with the corresponding computer software. Chromatograph "Chromos GC-1000" was equipped with packed and capillary columns. The packed column was used for the analysis of $H_2$, $O_2$, $N_2$, $CH_4$, $C_2H_6$, $C_2H_4$, $C_3$-$C_4$ hydrocarbons, CO, and $CO_2$, while the capillary column was used to analyze hydrocarbons. Temperature of the TC detector was 200 °C, the evaporator temperature 280 °C, and column temperature 40 °C. Carrier gas Ar velocity was 10 mL min$^{-1}$. A HP-PLOT Q capillary column, 30 m long and 0.53 mm in diameter, filled with polystyrene-divinylbenzene was used for analysis on an "Agilent Technologies 6890N" chromatograph.

The chromatographic peaks were calculated from the calibration curves plotted for the respective products using the "Chromos" software for pure substances (accurately measured quantities of the pure component or mixture of substances with known concentrations were injected into the chromatograph using a microsyringe). Based on the measured areas of the peaks, corresponding to the amount of the introduced substance, a calibration curve V = f (S) was constructed, where V is the amount of substance in mL, and S is the peak area in cm$^2$. Concentrations of the obtained products were determined on the basis of the obtained calibration curves. The balance of regulatory substances and products was ± 3.0%.

## 5. Conclusions

Catalysts prepared by SCS and incipient wetness methods based on $Co(NO_3)_2$-$Al(NO_3)_3$-urea systems were investigated in dry reforming of methane (model biogas). Analysis of the catalysts using XRD, SEM, TEM, and BET methods provided useful information in understanding the catalytic activity. The influence of the composition of initial components on formation of spinels, which were active in dry reforming of methane, was established. Temperature curves measured during SCS showed a second peak after SCS. This could only be explained by the reaction of carbon with metal oxides. Only reaction $Al_2O_3 + C \rightarrow Al + CO_2$ could explain the presence of Al in the product of reaction because hydrogen, which appears during the reaction, cannot reduce $Al_2O_3$ under conditions of synthesis. There are advantages of SCS catalysts in comparison with the catalysts prepared by traditional impregnation methods in dry reforming of methane. SCS is an express method (synthesis of the catalyst is carried out within a few minutes). This is an economical method (exothermic reaction is used for preparation of spinels at low temperatures). The synthesis of spinels is possible due to the formation of oxides from nitrates with high defect structure. These oxides can react with each other at much lower temperatures than oxides with well-formed crystal lattices. Because of exothermic reactions during SCS, the temperature rises by more than 1000 °C, the reaction rate is high, and the formed crystal lattice has a defective structure and is very active in catalysis. These conditions are also very suitable for tuning the crystal lattice of spinels and, accordingly, the activity of catalysts.

**Author Contributions:** Conceptualization, G.X. and S.A.T.; Methodology, G.X.; Validation, M.Z.; Formal Analysis, G.V. and D.Y.M.; Investigation, M.Z.; Writing—Original Draft Preparation, T.S.B.; Writing-Review & Editing, T.S.B.; Supervision, S.A.T.; Project Administration, T.S.B.; Funding Acquisition, S.A.T. All authors have read and agreed to the published version of the manuscript.

**Funding:** This research was funded by the Science Committee of the Ministry of Education and Science of the Republic of Kazakhstan (grant number AP08855562, AP08052090).

**Data Availability Statement:** The data presented in this study are available upon request from the corresponding author.

**Acknowledgments:** The authors are especially grateful to the staff of the laboratory of physical and chemical research methods.

**Conflicts of Interest:** The authors declare no conflict of interest. The funding sponsors had no role in the design of the study; in the collection, analyses, or interpretation of data; in the writing of the manuscript, and in the decision to publish the results.

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
