# Peer review of "Biogas Reforming over Al-Co Catalyst Prepared by Solution Combustion Synthesis Method"

_catalysts, doi:10.3390/catal11020274_

Round 1

Reviewer 1 Report

Summarizing what I read, there are a number of errors and overinterpretations, demonstrating a lack of knowledge, incredible in this paper. First, the XRD pattern of a mixture (up to six compounds!) does not allow to obtain an individual composition. A same line is assigned to 3 phases! Second, SEM technique does not allow to determine if the material is nano-scaled. Third, EDS analysis, which is conducted to get composition (not provided) cannot tell the presence of (crystalline) phases. The catalysis section is quite correct. But claiming that H2/CO ratio depends on the size of CH4 and CO2 molecules when space velocity is changed is alarming, as well as that CH4 conversion depends on the parameters of the crystal lattice of CoAl2O4 spinel (six phases present), “In CH4, the C–H bond length, which must be broken in accordance with the reaction mechanism, is 1.54 Å. In spinel, the size of the crystal lattice varies from 1.559 to 1.561 Å.”!!

This paper has certainly not been checked by a senior scientist. I ask for rejection.

Author Response

Reviewer 1:

Many thanks to the reviewer for critical comments and recommendations.

1.First, the XRD pattern of a mixture (up to six compounds!) does not allow to obtain an individual composition. A same line is assigned to 3 phases!

Positive identification was made in everyone of the six phases taking into account a) the known reactions taking place in this system and b) distinct peaks that were found in the XRD spectra. In addition, independent calibrations were carried out to be able to ascertain the presence of the phases even in cases when the peaks overlap.

2.Second, SEM technique does not allow to determine if the material is nano-scaled. Third, EDS analysis, which is conducted to get composition (not provided) cannot tell the presence of (crystalline) phases. The catalysis section is quite correct.

The SEM we used is FEI Quanta Inspect which has a resolution of about 3-4nm. This enabled very clear identification of the nano-structure of the various phases present.  Quantitative EDS analysis was used to identify many of the oxide phases present which confirmed the XRD observations.

3.But claiming that H2/CO ratio depends on the size of CH4 and CO2 molecules when space velocity is changed is alarming, as well as that CH4 conversion depends on the parameters of the crystal lattice of CoAl2O4 spinel (six phases present), “In CH4, the C–H bond length, which must be broken in accordance with the reaction mechanism, is 1.54 Å. In spinel, the size of the crystal lattice varies from 1.559 to 1.561 Å.”!!

As it is well known from theory of catalysis there is correlation between size of crystal lattice and bonds of molecules which must be broken in reaction. They must be close, but not the same because if they will be the same there will be good adsorption and bad desorption. Also catalysts crystal lattice produced by SCS has defect structure and parameters determined from XRD  could be slightly different.

Reviewer 2 Report

Comments enclosed. 

Author Response

Reviewer 2:

We thank the reviewer for the critical and constructive comments of our work.

Regarding the comments on the composition of the reaction mixture, we inform you that an unfortunate technical error has occurred. Correct composition of the initial reaction mixture - CO2:CH4 = 1:1. The corresponding correction was made in the text of the article.

Corrections on the L35, L45-46, L50-51, L53, L80, L94, L95, L127-128 were made in the text.

Paragraph 2.1.2: The text of the sentence ‘SCS catalysts are nano materials’ has been replaced by “Nanosized particles were determined in catalysts prepared by SCS method”. “и’ – “and”. We believe that a brief description of the SEM method at the beginning of section is quite appropriate.

Figure 7: The results were recorded in a very close time frame.

L160-162: No hydrogen was added to the reaction mixture. It was a mistake in the description. “The 1 : 1 ratio is very important for the Fischer Tropsch process…” has been replaced by “The 1 : 1 ratio is very important for the production of dimethyl ether, which can be used as a fuel”.

L170:  This sentence is correct because no hydrogen was added to the reaction mixture.

Figure 8: The temperature – 900oC. “и’ – “and”.

L190-193: “850 – 900 °C” has been replaced by “850 and 900 oC”, and a temperature of 900 oC has been added to the caption of Figure 10.

L197: ‘It was found that the values of the process parameters are quite close…’ has been replaced by “yields of H2 for the compared processes as well as yields of CO are quite close”

L201-202: It was decided to remove the description of the process and results of steam-carbon dioxide conversion of methane from the article and devote a separate article to this material.

L259: “certain amounts of nitrate salts are used” has been replaced by “pre-calculated amounts of nitrate salts were used”.

L286-287: Regarding the comments on the composition of the reaction mixture, we inform you that an unfortunate technical error has occurred. Correct composition of the initial reaction mixture - CO2:CH4 = 1:1. The corresponding correction was made in the text of the article.

L287-L230: We believe that the mention of dry methane reforming reactions does not contradict the general meaning of the narration in this section.

L295-297: “is used” has been replaced by “was used”. “A capillary column is used to analyze organic substances.” has been replaced by “The capillary column was used to analyse hydrocarbons.”

Reviewer 3 Report

The work is devoted to the actual topic of catalyst design for methane dry reforming. The manuscript reports some results on Co-alumina catalysts preparation, characterization and testing. The manuscript lies within the scope of the journal. The manuscript is “more or less” well-written and well-organized. The work suffers from the lack of discussion.  I think the work could be published in Catalysts after major queries will be answered:

  1. I recommend to better analyze the results paying special attention to the quantity of carbon formed over each sample per unit of time and mass of catalyst. It would be great to look for catalyst activation procedure and its structure under reaction conditions. Catalyst activation and reaction launch procedure is missed in Section 4. Materials and Methods.
  2. Comparison of lattice parameter and length of the chemical bonds in molecules is not obvious to be related to catalytic activity. It depends on the structure of adsorbed species. It would be useful to apply DFT or at least literature data.
  3. I recommend rewriting the Abstract and Conclusions. The main ideas and results have to be outlined. Sentence “Some advantage of SCS catalysts in comparison with catalysts prepared by the traditional impregnation method in the processes of dry reforming of methane was shown.” is not informative. The reader wants to know these advantages. The possibility of spinel formation is well-known. I think incipient wetness impregnation is more common than “impregnation of moisture capacity”.
  4. Table 1. Please provide any evidence on the occurrence of reactions of Al2O3 and CoO with carbon.
  5. SEM analysis. How did authors determine type of structures (cubic) using EDS?
  6. Fig. 7. Please add equilibrium values.

Author Response

Reviewer 3:

Many thanks to the reviewer for critical comments and recommendations.

  1. I recommend to better analyze the results paying special attention to the quantity of carbon formed over each sample per unit of time and mass of catalyst (In this article we study SCS with the same quantity of the reducer (organic substance), thus quantity of carbon which was produced during reaction was approximately the same. Carbon react with oxides (this is the only explanation why we have metals in the product. Thank you for advice,it will be taken into consideration during next studies). It would be great to look for catalyst activation procedure and its structure under reaction conditions (Thanks for the recommendation. Your proposal will be taken into account in the further research). Catalyst activation and reaction launch procedure is missed in Section 4. Materials and Methods. (Experiments on the partial oxidation of methane to synthesis gas were carried out on flow type installation at atmospheric pressure in a tubular quartz reactor with a fixed catalyst bed without any pre-reduction. Catalyst was placed in the central part of reactor and quartz wool placed above and below the catalyst bed. Changes have been made to the text of the article).
  2. Comparison of lattice parameter and length of the chemical bonds in molecules is not obvious to be related to catalytic activity. It depends on the structure of adsorbed species. It would be useful to apply DFT or at least literature data.

As it is well known from theory of catalysis there are different parameters influencing catalysts activity, one of them is correlation between size of crystal lattice and bonds of molecules which must be broken in reaction. They must be close, but not the same because if they will be the same there will be good adsorption and bad desorption. In this work we found some data proving correlation between those parameters.

  1. I recommend rewriting the Abstract and Conclusions. The main ideas and results have to be outlined. Sentence “Some advantage of SCS catalysts in comparison with catalysts prepared by the traditional impregnation method in the processes of dry reforming of methane was shown.” is not informative. The reader wants to know these advantages. The possibility of spinel formation is well-known. I think incipient wetness impregnation is more common than “impregnation of moisture capacity”.

Advantages of SCS in comparison with catalysts prepared by traditional impregnation method are: 1.SCS is express method (synthesis of catalyst is carried out within a few minutes). 2. This is an economical method (exothermic reaction is used for preparation of spinels at low temperatures). 3. The synthesis of spinels is possible due to formation of oxides from nitrates with high defect structure. These oxides can react with each other at much lower temperatures than oxides with well-formed crystal lattice. 4. Because of exothermic reactions during SCS, the temperature rises more than 1000 °C, the reaction rate is high, the formed crystal lattice has a defective structure and is very active in catalysis. Those conditions are also very suitable for tuning the crystal lattice of spinels and, accordingly, the activity of catalytsts. Yes, we are agree that the term “incipient wetness impregnation” is more common than “impregnation of moisture capacity”. Changes have been made to the text of the article.

  1. Table 1. Please provide any evidence on the occurrence of reactions of Al2O3 and CoO with carbon.

Temperature curves measured during SCS show a peak after SCS. This could be explained by the reaction of carbon with metal oxides. Unfortunately, we currently have such a picture for other conditions, so we cannot insert it into the article.

  1. SEM analysis. How did authors determine type of structures (cubic) using EDS?

Elemental analysis by EDS compared with XRD analysis and compositions of phases were confirmed. Structure was determined by SEM and XRD.

  1. Fig. 7. Please add equilibrium values. No

Round 2

Reviewer 2 Report

Based on your response and the corrections made, I recommend the paper to be published.

Author Response

Reviewer 2:

We would like to thank the reviewer for his constructive analysis and positive decision.

Reviewer 3 Report

I persevere authors to improve their manuscript in terms of discussion quality and application of modern level of catalysis consideration. For example, "correlation between size of crystal lattice and bonds of molecules which must be broken in reaction" is not direct and depends on the adsorption geometry over the surface. Why authors do not compare sqrt(2)*lattice parameter with bond length, which is more feasible for cubic lattice?

Considering response to reviewer I agree only with authors response for the 3rd question, except the sentence "that CoAl2O4 spinel formation is due to substitution of Al3+ with Co2+ cations". Formally, Co2+ cations occupy voids in the structure of Al2O3.

Other questions were ignored.

So, I still recommend major revision.

Author Response

Reviewer 3:

Many thanks to the reviewer for critical comments and recommendations.

  1. I recommend to better analyze the results paying special attention to the quantity of carbon formed over each sample per unit of time and mass of catalyst. In this article we study SCS with the same quantity of the reducer (organic substance), thus quantity of carbon which was produced during reaction was approximately the same. Carbon reacts with oxides which is the only possible explanation for the presence of metals in the product. Thank you for your comments anyway, which will be taken into consideration during next studies. It would be great to look for catalyst activation procedure and its structure under reaction conditions. Thanks for the recommendation. Your proposal will be taken into account in the further research. Catalyst activation and reaction launch procedure is missed in Section 4. Materials and Methods. Experiments on the partial oxidation of methane to synthesis gas were carried out in a flow type installation at atmospheric pressure in a tubular quartz reactor with a fixed catalyst bed without any pre-reduction. Catalyst was placed in the central part of reactor and quartz wool placed above and below the catalyst bed. These changes have been added to the text in the article.
  2. Comparison of lattice parameter and length of the chemical bonds in molecules is not obvious to be related to catalytic activity. It depends on the structure of adsorbed species. It would be useful to apply DFT or at least literature data.

As it is well known from theory of catalysis there are different parameters influencing catalysts activity, one of them is correlation between size of crystal lattice and bonds of molecules which must be broken in reaction. They must be close, but not the same because if they will be the same there will be good adsorption and bad desorption. In this work we found some data proving correlation between those parameters.

  1. I recommend rewriting the Abstract and Conclusions. The main ideas and results have to be outlined. Sentence “Some advantage of SCS catalysts in comparison with catalysts prepared by the traditional impregnation method in the processes of dry reforming of methane was shown.” is not informative. The reader wants to know these advantages. The possibility of spinel formation is well-known. I think incipient wetness impregnation is more common than “impregnation of moisture capacity”.

Advantages of SCS in comparison with catalysts prepared by traditional impregnation method are: 1.SCS is express method (synthesis of catalyst is carried out within a few minutes). 2. This is an economical method (exothermic reaction is used for preparation of spinels at low temperatures). 3. The synthesis of spinels is possible due to formation of oxides from nitrates with high defect structure. These oxides can react with each other at much lower temperatures than oxides with well-formed crystal lattice. 4. Because of exothermic reactions during SCS, the temperature rises more than 1000 °C, the reaction rate is high, the formed crystal lattice has a defective structure and is very active in catalysis. Those conditions are also very suitable for tuning the crystal lattice of spinels and, accordingly, the activity of catalysts. Yes, we agree that the term “incipient wetness impregnation” is more common than “impregnation of moisture capacity”. Changes have been made to the text of the article.

  1. Table 1. Please provide any evidence on the occurrence of reactions of Al2O3 and CoO with carbon.

Temperature curves measured during SCS show a peak(second peak) after SCS under all conditions. This can only be explained by the reaction of carbon with metal oxides. Only  reaction Al2O3 + C→Al + СО2can explain presence of Al in the product of reaction, because hydrogen, which appears during reaction can’t reduce Al2O3 under conditions of synthesis. This is shown in Figure 2 now included in the article.

Temperature curves during SCS of sample with initial batch 30% Cο(ΝΟ3)2 + 70% Al(NO3)3 + 60% urea

Temperature method describtion was added into Materials and Methods section.

  1. SEM analysis. How did authors determine type of structures (cubic) using EDS?

Elemental analysis by EDS was correlated with XRD analysis and compositions of the cubic phases were confirmed.

4.Fig. 7. Please add equilibrium values. No equilibrium values can be shown for these reactions since SCS gives metastable compounds.

5.We add comment according to reviewer comment about voids: The concentration of Al(ΝΟ3)3 plays an important role in the deformation of the crystal lattice. The higher the concentration of Al(ΝΟ3)3 in the initial solution, the more Co2+ ions (0.72 Ǻ) can replace the Al3+ ions (0.51 Ǻ) in the matrix, even though generally Co2+ cations occupy voids in the Al2O3 matrix, something which was not observed by XRD.

Correction in the article after 2 Round is marked in turquoise.
